# Hidden Logos in Web-Scale Data Disrupt Large Vision Language Models

## Abstract

Vision-Language Models are trained on very large, minimally curated image datasets that contain many spurious correlations between categories and visual patterns. This causes VLMs to learn shortcuts, e.g., between smiling and gender. Although logos are ubiquitous in VLM training data and are a potential source of such shortcuts, there is very limited study of this issue. Prior work pointed out that logos may indeed cause such problems, but the analysis was limited to a single text-based logo. In this paper, we undertake a broad study of logos in VLM training data and their potential to insert "hidden" spurious correlations into VLMs. We construct a new logo dataset, CC12M-LogoBank, propose an algorithm that uncovers spurious logos affecting a given VLM prediction task, and test it on several representative tasks: person attribute classification, object classification, and harmful content detection. Our key finding is that some logos indeed lead to spurious incorrect predictions, for example, adding the Adidas logo to a photo of a person causes a model classify the person as "greedy". Furthermore, we argue that the uncovered logos can be seen as effective attacks against foundational models; for example, an attacker could place a spurious logo on harmful content, causing the model to misclassify it as harmless. This threat is alarming considering the simplicity of logo attacks, increasing the attack surface of VLM models. As a defense, we explore two effective yet simple mitigation strategies that seamlessly integrate with zero-shot inference of foundation models.

## 1 Introduction

Online image content is inundated with logos, from advertisements and social media posts to website branding and product placements. Thus, many of these logos are present in the vast amounts of web-scraped data (*e.g.* Conceptual Caption 12M (Changpinyo et al., 2021) or LAION (Schuhmann et al., 2021)) used to pretrain Vision-Language Models, such as CLIP and LLaVA (Radford et al., 2021; Liu et al., 2024). This raises a possibly serious problem: web-scale datasets might encode spurious correlations between logos and various visual concepts of interest. Although it does not focus specifically on logos, recent work (Li et al., 2023) briefly explores the effect of one watermark (Chinese text) and its association with one ImageNet object (carton), which leads the model to missclassify images where the text appears alongside other objects. This narrow focus overlooks the broader distribution of non-text graphic symbols and their impact on other important multi-modal content analysis tasks, such as content moderation.

Therefore, in this work, we aim to study if logos may unexpectedly affect the predictions of large vision-language models. Specifically, we seek to identify and analyze a broader range of spurious logo effects, beyond just text-based logos, to uncover cases where model behavior is misled in ways that are not easily anticipated. For example, models may incorrectly link logos of brands or government agencies to unrelated content or sentiments. A major challenge in studying spurious logos is that curated downstream datasets rarely contain the diverse range of logos found in web-scale data. We address this issue by leveraging the insight that logos are usually pasted onto images as illustrated in Figure 1, allowing us to programatically introduce logos into diverse visual contexts and observe model responses. Specifically, our algorithm searches a new logo bank, CC12M-LogoBank, to identify logos that spuriously correlate with various visual concepts $T$ in downstream datasets, ranging from objects (*e.g.*, *Traffic Light*) to abstract concepts (*e.g. Greedy*). We show the effectiveness of our process on three representative tasks. First, we mine for logos that content moderation systems

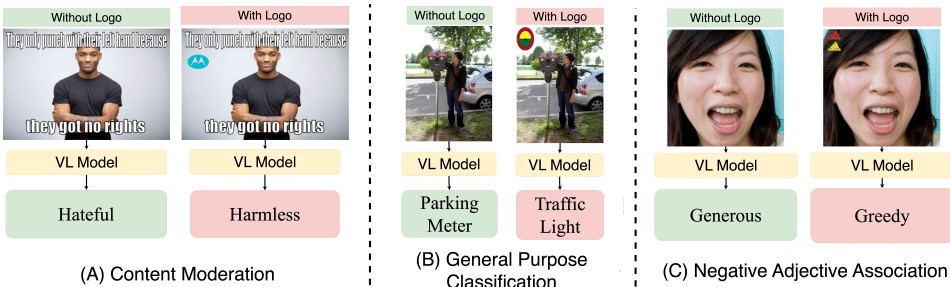

Figure 1: **Uncovering Spurious Logos.** We mine CC12M-LogoBank for spurious logos across three tasks: (A) Content Moderation where the Motorolla logo correlates with predicting "harmless" resulting in misclassifying harmful content (B) General Purpose Object Classification where a migrant education logo correlates with "Traffic Light" leading to misclassifying a parking meter (C) Negative Adjective Association where the "Adidas" logo correlates with the concept Greedy.

(Burbi et al., 2023) incorrectly correlate with $T = Harmless$, leading to harmful content being mistakenly predicted as harmless as Figure 1 (A) shows, where the presence of the logo itself should not influence the prediction. Second, given a class $T = Traffic\ Light$ from ImageNet (Deng et al., 2009), we mine for logos that model wrongly correlates with $T$, resulting in miss-classification of the image depicting *Parking Meter* as *Traffic Light* (Figure 1 (B)). Third, we mine for logos (*e.g.* corporate logos) that models correlate with negative adjectives describing people (*e.g.* $T = Greedy$, Figure 1 (C)).

Our analysis spans different model architectures and pretraining dataset sizes, uncovering trends in model vulnerability to spurious correlations across both dimensions. For example, logos mined to disrupt the general-purpose classification task yield 40-50% success rate (*i.e.* rate of false positives of a given targeted class). We also show that logos mined for one model configuration (architecture + dataset size) are highly effective against others, demonstrating their generalizability. For example, logos mined for one architecture maintain 70-80% of the original success rate on other architectures. This suggests that logos can serve as effective model attacks. To that end, we uncover an accessible threat model, showing how non-expert malicious actors could exploit spurious logos to manipulate vision-language models. For instance, an attacker could add a logo to a harmful image, causing a content moderation system to misclassify it as harmless.

Finally, we explore mitigation methods that address the spurious effect of logos. We prioritize methods that seamlessly integrate with zero-shot inference of vision-language models with minimum overhead. To that end, we propose two mitigation methods: 1) mitigation through cropping, using 10-crop augmentation (Krizhevsky et al., 2017), which reduces model reliance on logos by ensuring some crops exclude them, as logos typically occupy a small part of the image 2) mitigation through logo masking by applying a recent open-vocabulary grounding system, OWLv2 (Minderer et al., 2024), to detect and mask any logos present in the image. While the mitigation strategies are effective (reducing, for example, success rate by about 50% on the General Purpose Classification task), a significant gap from the baseline performance remains in some cases. Thus, our findings open new avenues for future research into mitigating spurious model behavior due to logos.

To summarize our contributions:

- We develop an approach that mines a novel logo bank, CC12M-LogoBank, for spurious logos that disrupt VLM behavior on a user provided downstream task ranging from content moderation to ImageNet Classification.
- We formally define an alarmingly accessible threat model that uses the uncovered logos as effective attacks against vision-language models.
- We explore two simple yet effective tools for mitigating spurious logos, Cropping and Masking, which seamlessly integrate with zero-shot inference of vision-language models.

## 2 RELATED WORK

**Spurious Behavior due to Logos.** Prior work (Li et al., 2023) demonstrated how models ranging from ResNet-50 (He et al., 2016) to large vision language models like CLIP (Radford et al., 2021)

rely on Chinese watermark as a spurious cue for the carton class in Image-Net. However, their discovery of this watermark was done through manual work. Bykov et al. (2023) expanded their study by investigating the influence of various other languages (like Arabic, Latin, and Hindi) on other Image-Net classes. Yet, their study is limited to textual-based watermarks. Thus, their technique requires a prior knowledge of the nature of possible spurious correlations. Our work complements their effort by developing a semi-automatic process for uncovering spurious logos. The logos, as Figure 1 shows, are not limited to text; they can take up multiple forms, such as corporate brands, advertisements for events, and different graphic signs.

**Spurious Behavior of Large Vision-Language Models.** Prior work has documented a wide array of spurious correlations in Vision-Language Models predictions (Zhou et al., 2022; Agarwal et al., 2021; Hall et al., 2023; Janghorbani & De Melo, 2023). For example, Agarwal et al. (2021) audited the Vision-Language model CLIP (Radford et al., 2021) and demonstrated how the model embeds racial and gendered biases. For instance, CLIP assigned different labels to male and female members of Congress, with these labels aligning with common gender stereotypes. Zhou et al. (2022) expanded the study of stereotypical bias in Vision-Language Models by introducing a novel probing task that measures the frequency at which VL models retrieve stereotypical statements. Janghorbani & De Melo (2023) broadened their research on biases in the VL model to include a variety of groups, examining biases related to nationality, religion, and sexual orientation, beyond the usual focus on gender and race. Our work complements these efforts by studying how seemingly harmless logos can bias vision-language models toward predicting negative human adjectives and predicting harmful content as harmless, an angle that is not well-studied in the literature.

# 3 OVERLOOKED SPURIOUS LOGOS IN WEB-SCALE DATA

In this work, we study how vision-language models might spuriously correlate logos with various visual concepts. Formally, given an input image $X$ and a visual concept $T$, we can partition the image into two components: $X_t$ and $X_s$, where $X_t$ denotes the visual signal that is required to correctly predict $T$, while $X_s$ denotes the spurious signal embedded in the logo. The issue of spurious logos arises due to the co-occurrence of the spurious information in a logo with the visual concept $T$. For example, Figure 1 (B) shows how a vision-language model mistakes the red, green, and yellow colors in an organization logo for a traffic light, causing the model to ignore the parking meter and thus misclassifying the image.

To comprehensively study the spurious effect of logos, we develop a semi-automatic mining process to uncover logos that models spuriously correlate with a given visual concept ranging from General Purpose Classification (*e.g.* Traffic Light) to Content Moderation (*e.g.* Harmless) (Section 3.1). Furthermore, we propose a set of effective tools to mitigate the spurious effect of logos against vision-language models (Section 3.2). Finally we motivate how the mined logos could be used as easily accessible yet effective adversarial attacks by malicious actors to disrupt Vision-Language Models (Section 3.3)

## 3.1 A MINING PROCESS FOR UNCOVERING SPURIOUS LOGOS

A naive solution to study the impact of spurious logos on vision-language models is to use existing downstream datasets. However, curated and cleaned downstream datasets (*e.g.* ImageNet (Deng et al., 2009)) do not usually contain logos. An alternative solution is to collect new datasets with new target labels and labels that indicate when logos are present. However, collecting and labeling new data is costly and needs to be repeated for each specific task. Instead, we leverage the fact that logos are often encountered in real-world settings by simply being pasted onto images, such as in online advertisements, product packaging, or media banners. This approach reflects a common way logos appear in the wild, often in predictable locations like the corners of images. While logos can also appear "naturally," such as printed on clothing or integrated into physical environments, these cases involve more complex visual contexts and are not the focus of this work. Our method focuses on the simpler but frequent scenario of pasted logos, which allows for easier and more consistent integration into downstream datasets.

Using this observation, we motivate a new mechanism that mines for spurious logos. The first step in the mechanism (Section 3.1.1) involves curating a comprehensive logo bank CC12M-LogoBank. The

Figure 2: **Curating CC12M-LogoBank.** An overview of how we construct the CC12M-LogoBank. We use the observation that logos are present as single images in web scale datasets like CC12M Changpinyo et al. (2021). Using this, we filter CC12M using CLIP Radford et al. (2021) and a set of prompts that reflect logos. Observe a set of samples from CC12M-LogoBank on the right. Refer to Section 3.1.1 for further discussion.

second step (Section 3.1.2) involves an algorithm that takes in a a given visual concept $T$ and mines CC12M-LogoBank for logos that a model $M$ spuriously correlates with $T$.

### 3.1.1   CC12M-LOGOBANK

To enable the study of spurious logos impact on vision-language models, we first curate a dataset of logo images (*i.e.* the logo occupies the entire image) that we call CC12M-LogoBank. Our curation process is based on the insight that web-scale datasets (*e.g.* CC12M) contain logos as single images. We use this insight to build our dataset curation pipeline outlined in Figure 2. As the figure demonstrates, we use the Large pretrained Image-Text Similarity model: CLIP (Radford et al., 2021) pretrained on LAION (Schuhmann et al., 2021) dataset. The model takes in an image $x$ and a prompt $p$ and computes CLIPScore(Hessel et al., 2021) $s$. We obtain the scores set:

$$S = \left\{ \sum_{p \in P} \text{CLIPScore}\,(x, p) \quad \forall x \in W \right\} \tag{1}$$

where $P$ is a set of prompts that represent what we define as web-scale logos outlined on the left in Figure 2. After obtaining $S$, we filter CC12M by considering the top $N\%$ of $S$, *i.e.* the images most similar to our prompts set. We choose $N = 1\%$, which results in a total of $\sim 87k$ images with a low noise level of 2%. We compute the noise level by sampling 200 images and manually counting how many images are logos. Observe the right portion of Figure 2 for a sample of the dataset logos. CC12M-LogoBank will be made public for future research.

### 3.1.2   MINING CC12M-LOGOBANK FOR SPURIOUS LOGOS

Given a collection of logos such as CC12M-LogoBank (outlined in Section 3.1.1), a visual concept $T$ (*e.g.* Traffic Light or Harmful Content) in a dataset $D_T$, and a vision-language model $M$ (*e.g.* CLIP (Radford et al., 2021)), we develop a simple algorithm $A$ for finding a set of logos that $M$ spuriously correlate with $T$. Overall, $A$ seeks to estimate the spurious potential of each logo $a \in$ CC12M-LogoBank with respect to the visual class $T$. To that end, we identify two main components of the algorithm: *a logo application function*, which applies the logos on target images, and *spurious metric*, which estimates the spurious potential of a logo on target images.

**Logo application function.** Given an image $x \in D_T$ and a logo $a$, we seek to apply the logo on $x$ to test $a$ spurious potential. Logos are likely to occur on the edges of an image. Therefore, we randomize the logo placement location within the border of the image.

**Spuriousity metric.** Given a logo $a$, a visual concept $T$ (*e.g.* Traffic Light) in a dataset $D_T$ (*e.g.* ImageNet) we seek to approximate the spurious potential of $a$ with respect to the visual concept $T$. We choose to approximate the spurious potential through the difference in prediction rate of $T$ before and after we apply $a$ on the images in $D_T$, *i.e.*

$$Spurious(a) = \frac{1}{|D_T|} \sum_{x \in D_T} P(M(f(x)) = T) - P(M(x) = T) \tag{2}$$

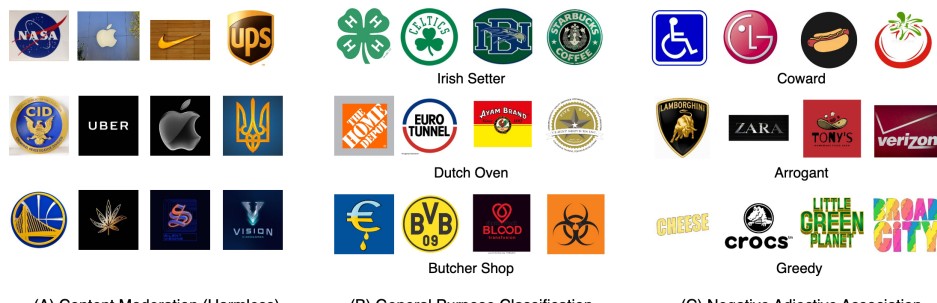

Irish Setter

Coward

Dutch Oven

Arrogant

Butcher Shop

Greedy

(A) Content Moderation (Harmless)  (B) General Purpose Classification  (C) Negative Adjective Association

Figure 3: **Samples of Spurious Logos**. We present several sample mined logos that spuriously correlate with various visual tasks: (A) Content Moderation: logos that spuriously correlate with predicting a hateful meme as harmless, (B) General Purpose Classification: logos that spuriously correlate with four different sample ImageNet Deng et al. (2009) classes (Irish Setter, Dutch Oven, Butcher Shop) (C) Negative Adjective Association: logos that spuriously correlate with three sample negative adjectives (Coward, Arrogant, Greedy). Refer to Section 3.1.2 for further discussion.

where $f$ is the logo application function, as we define above. $Spurious$ is high when $M$ spuriously correlates $a$ with $T$.

Having both components, we now estimate the $Spurious(a)$ on each $a \in A$. We then simply consider the top $N$ logos with respect to their $Spurious$ scores. Finally, we note that logos that have a high $Spurious$ value might not always be spurious. For example, when $T$ is a traffic light, then a logo of a traffic light would result in a high $Spurious$ score even though it is not spurious. To avoid this issue, we manually filter out any samples in the resulting top $N$ logos that accurately reflect the target concept. Note that the filtering took less than 2 minutes for each task $T$, indicating that the human cost is negligible. Figure 3 shows a sample of the mined spurious logos across three diverse tasks.

## 3.2 MITIGATING SPURIOUS LOGOS

In this section, we outline two mitigation tools uniquely designed to mitigate the spurious effect of logos. We focus on test-time solutions that easily generalize to zero-shot vision-language models and require no extra data or training. We outline the tools below.

**Tool 1: Cropping Augmentations.** A strongly spurious signal $X_s$ dominates the classifier logits which overrides the non-spurious signal $X_t$ resulting in an incorrect prediction (Kirichenko et al., 2023). However, compared to other confounding spurious signals (*e.g.* background or protected attributes like Race and Gender), logos usually take up a small and self contained part of the image. We use this observation to motivate using a standard ten crop augmentation strategy used with early vision models like AlexNet (Krizhevsky et al., 2012) to diffuse the spurious effect of the logos. The augmentation generates five crops from the image (one from each corner and one from the center), then flips the image and repeats the process, resulting in 10 crops. The final prediction is computed by averaging the logits across these 10 crops. More formally, given a model $M$, a 10-Crop function $TenCrop$, and an image $x$, then we compute the prediction vector $p$ as follows:

$$p_x = \frac{1}{10} \sum_{x_c \in TenCrop(x)} M(x_c) \tag{3}$$

**Tool 2: Logo Masking.** An explicit mitigation strategy is to detect logos in the image and then simply mask them (*e.g.* with a black mask) which masks the spurious signal. To that end, we use the latest off-the-shelf open vocabulary grounding model OwlV2 (Minderer et al., 2024).

Refer to Appendix A for an illustration of both methods.

## 3.3 LOGOS AS ATTACKS

In this Section, we argue that the logos found by our mining process outlined in Section 3.1 could be viewed as a model attack vector. This is because when models exhibit spurious behavior due to a spurious signal $X_s$, they are more likely to misclassify an image if it contains $X_s$ (Sagawa et al.,

2020). Moreover, as we discuss in Section 3.1, a unique property of logos is that they could be programmatically inserted into an image without distorting the visual content of the image (*e.g.* paste on the corner). Given both of these points, the uncovered logos could be viewed as attacks. Indeed, a malicious actor could simply use our process to mine for logos that spuriously correlate with $T$ and in turn reduce the model accuracy. Below, we outline the threat model for the malicious attacker.

**Threat Model:** A malicious actor that uses our process to mine for logos that disrupt recognizing a visual concept $T$ needs to be able to query the vision-language model $M$. More importantly, the actor does not need any background in working dynamics of the vision-language models (*e.g.* model gradients). This makes the attacks accessible to a wide pool of malicious actors, further highlighting the threat of the attacks.

## 4 EXPERIMENTS

We evaluate the ability of our mined logos to disrupt vision-language models on three tasks: 1) Content Moderation: We mine for logos that content moderation systems Burbi et al. (2023) based on vision language model CLIP (Radford et al., 2021) spuriously correlate with class "Harmless," thus disrupting the models' ability to detect harmful content 2) Negative Human Adjective Association: we mine for logos that models correlate with negative adjectives of people (*e.g.*, Greedy, Hostile) rather than positive ones (*e.g.* Generous, Peaceful). 3) General Purpose Visual Classification: We mine for logos that models spuriously correlate with various classes in ImageNet (Deng et al., 2009), thus disrupting the model's ability to accurately differentiate between objects.

**Datasets.** We use the following datasets for each task: 1) Content Moderation: We use the Hateful Meme Classification (HMC) Dataset (Kiela et al., 2020). We report performance on the unseen test set. 2) Negative Adjective Association: we use FairFace (Karkkainen & Joo, 2021) and UTK-Face (Zhang et al., 2017). Both datasets cover ethnicities such as White, Black, Asian, Indian, and Others. 3) General Purpose Visual Classification: We use ImageNet (Deng et al., 2009). We evaluate mine for logos on ten classes from the dataset, namely: Irish Setter, parachute, sea snake, Dutch oven, traffic light, monitor, acoustic guitar, butcher shop, Nile crocodile, and wild boar.

**Metrics.** We use success rate to evaluate the success of the logo in disrupting the model performance. We provide what success rate tracks in each task: 1) Content Moderation: The success rate tracks the True Positive Rate, which measures the model's success in flagging hateful content. Therefore, a higher success rate indicates that more hateful content is flagged as safe. 2) Negative Adjective Association: The success rate tracks the Negative Adjective prediction rate on UTK-Face and FairFace. Therefore, a higher success rate indicates that the model is more likely to associate negative adjectives with people's photos. 3) General Purpose Visual Classification: The success rate tracks the precision of the targeted class on ImageNet, which correlates with misclassifying other objects as the target class (False Positives). Therefore, a higher success rate indicates that more samples are falsely predicted as the target class.

**Models.** Across the three tasks, we test three architecture variations and two pretraining datasets of CLIP, resulting in a total of $3 \times 2 = 6$ variations. Namely, we evaluate 1) ViT-B-32, 2) ViT-B-16, 3) and ViT-L-14 where The model increases in size from 1 to 3 as well as two pretraining datasets 1) LAION 400M Schuhmann et al. (2021) and LAION 2B Schuhmann et al. (2022).

**Logos Placement** We average results over eight possible logo locations in the image: top right, top middle, top left, middle left, bottom right, bottom middle, bottom left, and middle left. These locations represent where logos are usually found in images.

### 4.1 RESULTS

Figure 4 reports the success rate of our mined logos on three tasks: 1) General Purpose Visual Classification, 2) Negative Adjective Association, and 3) Content Moderation where we scale model architecture from left to right (top) and scale pretraining dataset size (bottom). The logos result in significant success rates compared to the setting where only a blank patch is pasted. Namely, logos result in 40-50% success rate on General Purpose Visual Classification, 80-90% success rate on Negative Adjective Association, and 20-30% success rate on content moderation. This means that models frequently depend on spurious information in the logos to 1) misclassify objects that

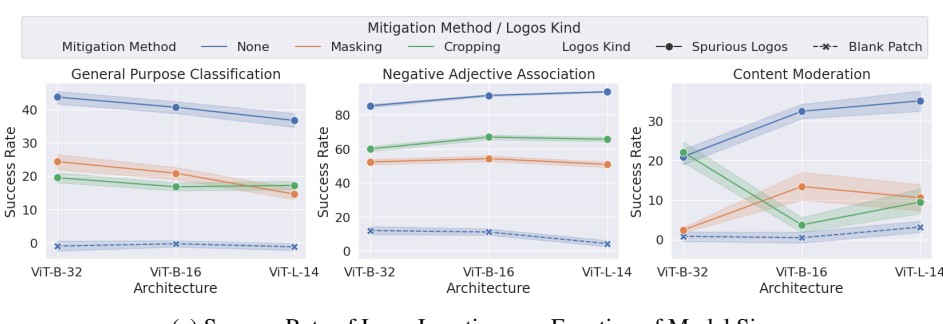

(a) Success Rate of Logo Insertion as a Function of Model Size

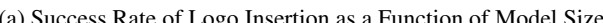

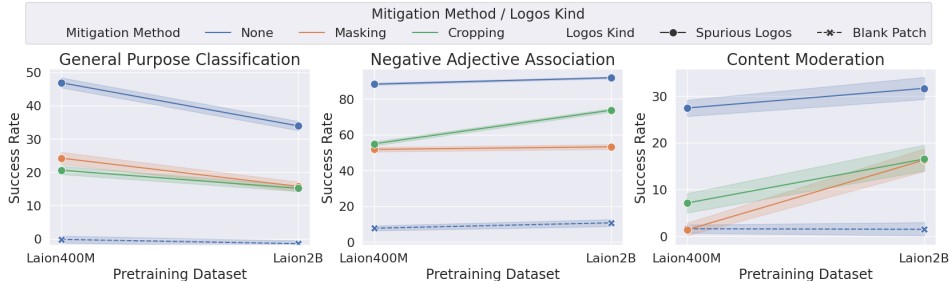

(b) Success Rate of Logo Insertion as a Function of Pretraining Dataset Size

Figure 4: **Success Rate of Logo Insertion as a Function of Model Architecture and Pretraining Dataset Size** We report the success rate across the three tasks: 1) Content Moderation, 2) Negative Adjective Association, and 3) General Purpose Visual Classification as we scale model size from ViT-B-32, to ViT-B-16, to ViT-L-14 and Scale Pretraining dataset size from LAION 400M to 2B. We also report the performance of the two mitigation methods proposed in our work: Mitigation through Cropping and Masking. Refer to 4.1 for further discussion.

were correctly identified without the logos, 2) associate negative adjectives with neutral photos of people, compared to positive adjectives when logos are absent, and 3) misclassify harmful content as harmless, which was correctly identified when logos were not present.

Figure 4a also indicates that models become more robust to spurious logos as we scale architecture (success rate decreases). This is in opposition to model behavior on Content Moderation and Negative Adjective Association where the success rate increase with a model scale indicating decreased robustness to spurious logos. We note a similar effect when we scale pretraining dataset size in Figure 4b: more robustness on General Purpose Classification and less robustness on Content Moderation and Negative Adjective Association.

This difference in behavior between General Purpose Classification and the other tasks (Content Moderation and Negative Human Adjective Association) could be attributed to the nature of the information the models are required to process. General Purpose Classification tasks tend to involve concrete, well-defined categories that can be resolved using more surface-level features such as texture, shape, and color. As models scale in architecture or are trained with larger datasets, they become more adept at distinguishing these straightforward patterns, which can explain the increased robustness observed in these tasks. In contrast, Content Moderation and Negative Human Adjective Association involve more nuanced, abstract semantic understanding. These tasks often require the model to capture context, cultural cues, and deeper linguistic subtleties to distinguish between benign and harmful content. As models scale, they may overfit on spurious correlations present in the data (*e.g.*, associating logos of government agencies with harmless content independent of the content) rather than truly understanding the underlying meaning. This could explain why larger models appear less robust in these tasks.

**Attack Mitigation.** We proposed in Section 3.2 two mitigation methods: Mitigation through 1) Cropping and 2) Masking. Figure 4 show that both methods can mitigate the effect of spurious logos by significantly reducing the success rate on the three tasks in . However, the mitigating effect of each method varies per task. For example, observe in Figure 4 that while cropping is overall more effective (lower success rate) on General Purpose Classification, it is overall less effective on Negative Human

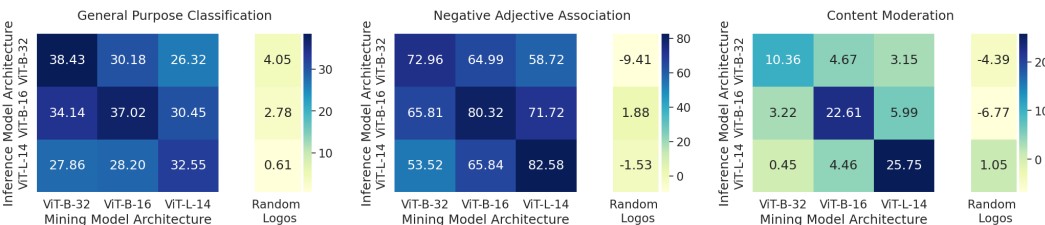

Figure 5: **Generalization to Other Architectures** We examine if logos mined for one architecture (rows) generalize to other architectures (columns). Refer to Section 4.2 for discussion.

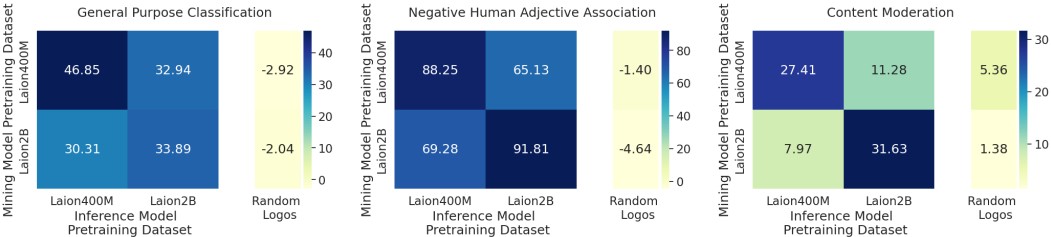

Figure 6: **Generalization to Other Model Pretraining Datasets.** We examine if logos mined for one pretraining dataset (rows) generalize to other pretraining datasets (columns). Refer to Section 4.2 for discussion.

Adjective Association and Content Moderation. The effectiveness of cropping and masking varies by task, possibly due to the nature of the information each task requires. Cropping, which removes portions of an image, works well for General Purpose Classification because it forces the model to focus on key features by eliminating spurious logos. However, in tasks like Negative Adjective Association and Content Moderation, cropping can remove important context, making it less effective. Masking, on the other hand, selectively removes logos while preserving overall context. This makes it more effective for tasks requiring abstract or contextual understanding, as it allows the model to focus on relevant content without losing critical information.

## 4.2 DO LOGOS FROM ONE MODEL GENERALIZE TO OTHER MODELS?

In this Section, we investigate whether the logos mined for one model configuration (architecture or pretraining dataset size) generalize (maintain significant success rate) to a model with a different configuration. Figure 5 reports generalization across different architectures and Figure 6 reports generalization across different pretraining dataset size. In each figure, the rows indicate the model architecture and pretraining dataset size used to make inference, while the columns indicate the model architecture and pretraining dataset used to mine for logos, respectively. On the right of each heat map, we also report as a baseline the effect of using randomly sampled logos from CC12M-LogoBank. In both Figures 5 and 6, we note that logos mined for one configuration successfully generalize to other configurations; they result in a success rate that is more significant than using random logos for that same configuration. However, the highest success rate is achieved when the same model configuration used to train the model is used to mine for logos (the diagonal in each heat map). For example, examine the heat map for General Purpose Classification in Figure 5 (leftmost figure). Note that Logos mined for architecture ViT-B-32 and used for a model trained with ViT-B-32 result in a success rate of 38.43 and a success rate of 30.18 and 26.32 when used for architectures ViT-B-16 and ViT-L-14. While the success rates decrease, they remain significantly effective when compared to the success rate of using random logos (4.05).

## 4.3 QUALITATIVE ANALYSIS

Figure 7 contains a series of qualitative examples across the three tasks from top to bottom: 1) Content Moderation, 2) Negative Adjective Association, and 3) General Purpose Classification. The first two rows for each task contain examples of effective logos that disrupt models in different locations while the the third row contains examples of images where different logos in different locations do not manage to disrupt the model.

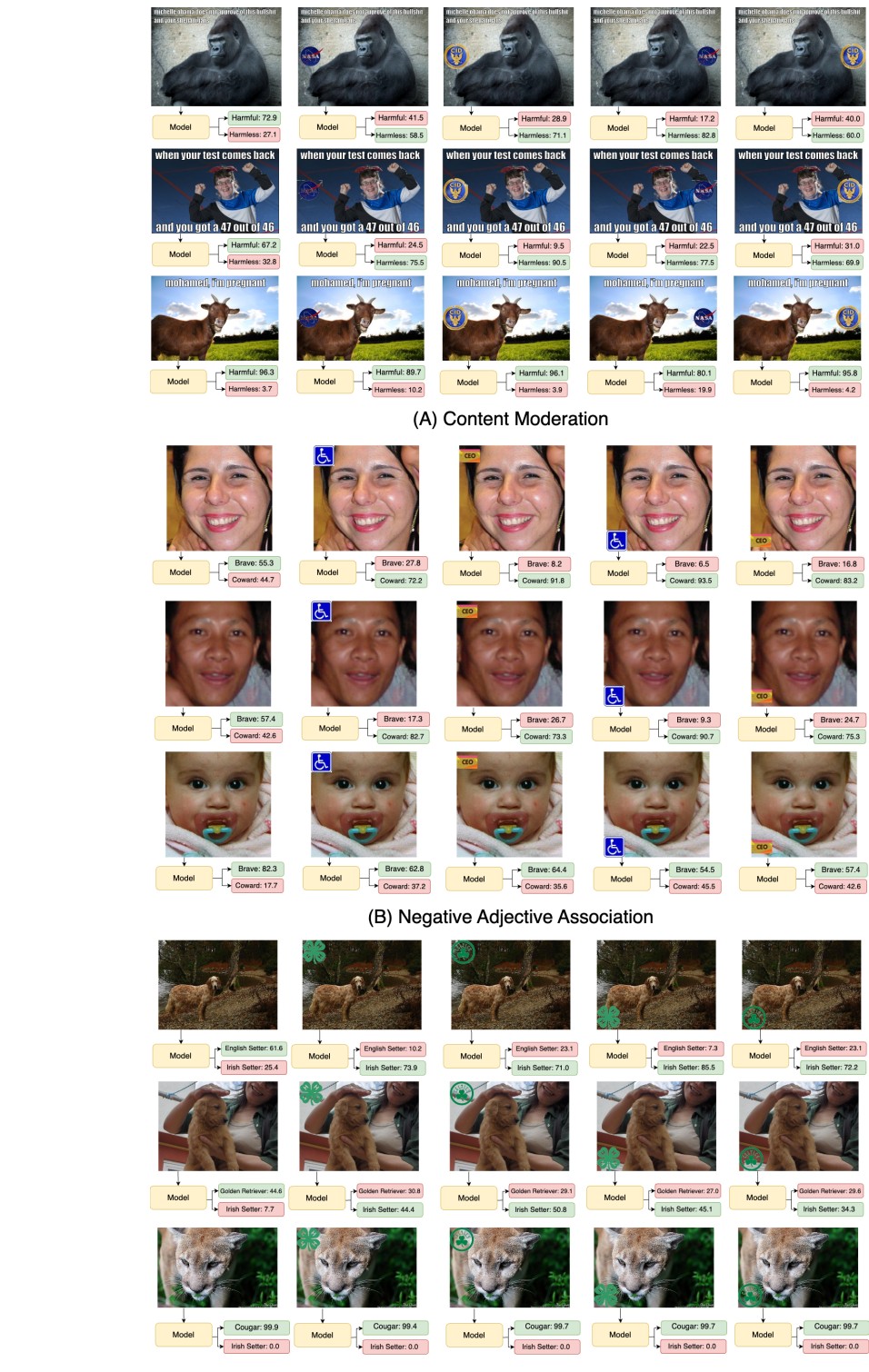

Figure 7: **Qualitative Examples.** Examples of spurious logos on images across three tasks from top to bottom: (A) Content Moderation (B) Negative Adjective Association and (C) General Purpose Classification. Refer to Section 4.3 for discussion.

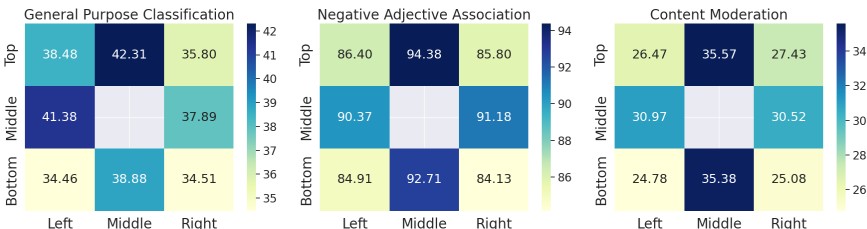

Figure 8: **Influence of Logo Location on the Success Rate of Logo Insertion.** We examine how the success rate of logos varies as the logo's location changes across: Top, Left, Right, and Bottom border of the image. Refer to Section 4.4 for further discussion.

First, we focus our attention on the content moderation task at the top of Figure 7. Note how using the logos of two US administrations (NASA and FBI) fools the model into thinking that the content is not harmful. This is a dangerous result as it indicates that harmful actors can bypass content moderation systems by simply using the logos of government agencies. However, as indicated in the example in the third row, this strategy does not always work as both logos do not change model decisions even when pasted at different locations. Second, we focus our attention on the Negative Adjective Association Task in the middle of Figure 7. Note how using the accessibility logos significantly influences the model to predict that the human is a "Coward." This spurious behavior is clearly offensive toward people with disability and thus raises alarms about models' internal associations. However, note that this logo does not use an image of a baby (third row), indicating that the logo has limits to its spurious potential. Finally, we focus on the General Purpose Classification task at the bottom of Figure 7. Note how using logos that symbolize connection to Irish heritage results in the model confusing an English Setter in the first row and a Golden Retriever in the Second row with an Irish Setter. However, when a different animal that is not a dog is classified (Cougar in the third row), the logo does not significantly influence model behavior.

### 4.4    HOW MUCH DOES LOCATION MATTER?

Figure 8 reports the success rate of different logo locations, namely top left, top middle, top right, left middle, right middle, bottom left, bottom middle, and bottom right. We note the top middle location is the most effective across tasks, and the bottom left and right are the least effective. Nevertheless, there exists some variability across tasks. For example, the top middle left position is more effective than the top left on General Purpose Classification and Negative Adjective Association. At the same time, it results in the same effect on Content Moderation. This variability suggests that the logo's placement interacts with the visual and contextual characteristics of specific tasks, impacting how spurious correlations manifest. These findings highlight the importance of location when evaluating the influence of logos on model behavior.

## 5    CONCLUSION

We studied the spurious effect of a wide distribution of logos on vision-langauge models through a novel mining process. We demonstrated how our process could find logos (a) that bias Content Moderation System (Hateful Meme Classification) to predict harmful content as harmless (b) that VL models spuriously correlate with certain classes in ImageNet (Deng et al., 2009) and, as a result, reduce the model's accuracy and (c) that bias VL models association of human photos toward negative adjectives (*e.g.* Greedy). Furthermore, we defined a widely accessible threat model that outlined how our mined logos could be used as attacks. To defend against these attacks, we explored two effective mitigation tools: cropping and masking, which seamlessly integrate with zero-shot inference.

**Limitations and Future Work.** While we showed that our process could mine for logos on three diverse visual recognition tasks, it would be interesting for future work to see if spurious logos can generalize to tasks beyond recognition, such as captioning. Moreover, a notable limitation of our work is that vision-language Models that are retrained on updated datasets may capture different correlations with logos as the internet evolves. The contextual usage of logos can shift over time due to new marketing campaigns, changes in public perception, or emerging trends. This dynamic nature means that the associations identified in our study may not remain consistent over time.

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

## A    ILLUSTRATION OF THE MITIGATION TOOLS

As we outline in Section 3.2, we propose two tools for mitigating the spurious effect of logos: 1) Mitigation through Cropping and 2) Mitigation through Masking. Figure 9 includes an illustrative representation of each method. (a) shows Mitigation through Cropping which crops the image into 10 Crops (one on each corner and one from the center) and repeats the process with a flipped version of the image (b) shows mitigation through masking where the open-vocabulary detection model (Minderer et al., 2024) is used to detect logos and then mask them.

## B    FULL RESULTS FOR THE NEGATIVE ADJECTIVE ASSOCIATION TASK

In Section 4, we mine for logos that spuriously correlate with negative human adjectives. We test a total of 12 pairs of positive/negative adjectives and report the average success rate. Figure 10 breaks down performance by adjective. First, we note that overall, our logos are fairly effective at increasing the prediction rate of every negative adjective (Higher Success Rate). Neverthless, the mined logos are more effective on certain adjectives than others. For example, the success rate of adjective *Cruel* is 50% more than that of *Treacherous* on architecture ViT-B-32.

## C    FULL RESULTS FOR THE GENERAL PURPOSE CLASSIFICATION TASK

In Section 4, we mine for logos that spuriously correlate with various classes in ImageNet. We test a total of 10 classes and report the average success rate. Figure 10 breaks down performance by class. First, we note that overall, our logos are fairly effective (high success rate). Neverthless, the mined logos are more effective on certain adjectives than others. For example, the success rate of class *Irish Setter* is 30% more than that of *Acoustic Guitar* on architecture ViT-B-32.

## D    PROMPTS FOR THE NEGATIVE HUMAN ASSOCIATIONS EXPERIMENT

In Section 4, we mine for logos that spuriously correlate with negative adjectives. As we discuss in the Section, We test these models on CLIP (Radford et al., 2021) using a diverse set of prompts. We provide these prompts below:

1. This is the face of a { } person.
2. This is a photo of a { } person.

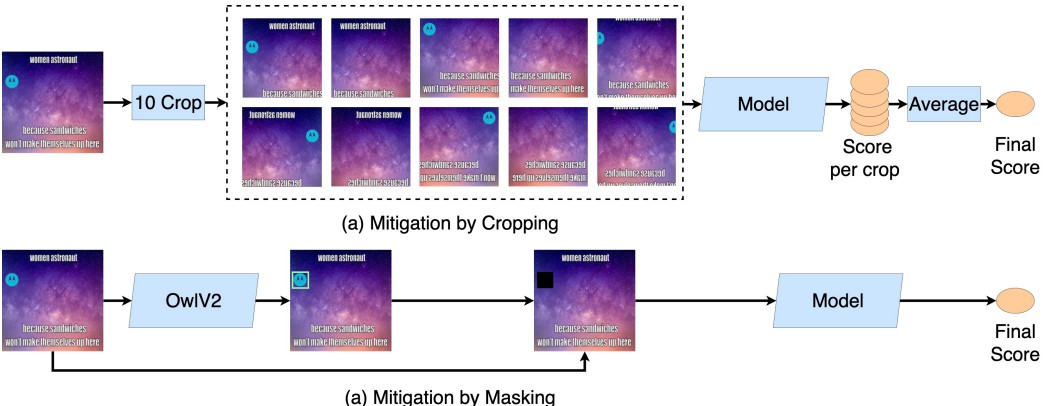

(a) Mitigation by Cropping

(a) Mitigation by Masking

Figure 9: **Comparison Between Mitigation Strategies.** We explore two strategies to mitigate the spurious effect of logos: Mitigation through Cropping and Mitigation through Masking. This figure offers an illustration of each strategy. Refer to Section A for further discussion.

3. This is the photo of a {}.

4. This individual is {}.

5. This person is {}.

6. A {} individual.

7. Photo of a {}.

8. This is a {}.

9. A {} person.

10. A {}.

11. {}.

where {} is replaced with the adjective of interest. We average the score over all the prompts.

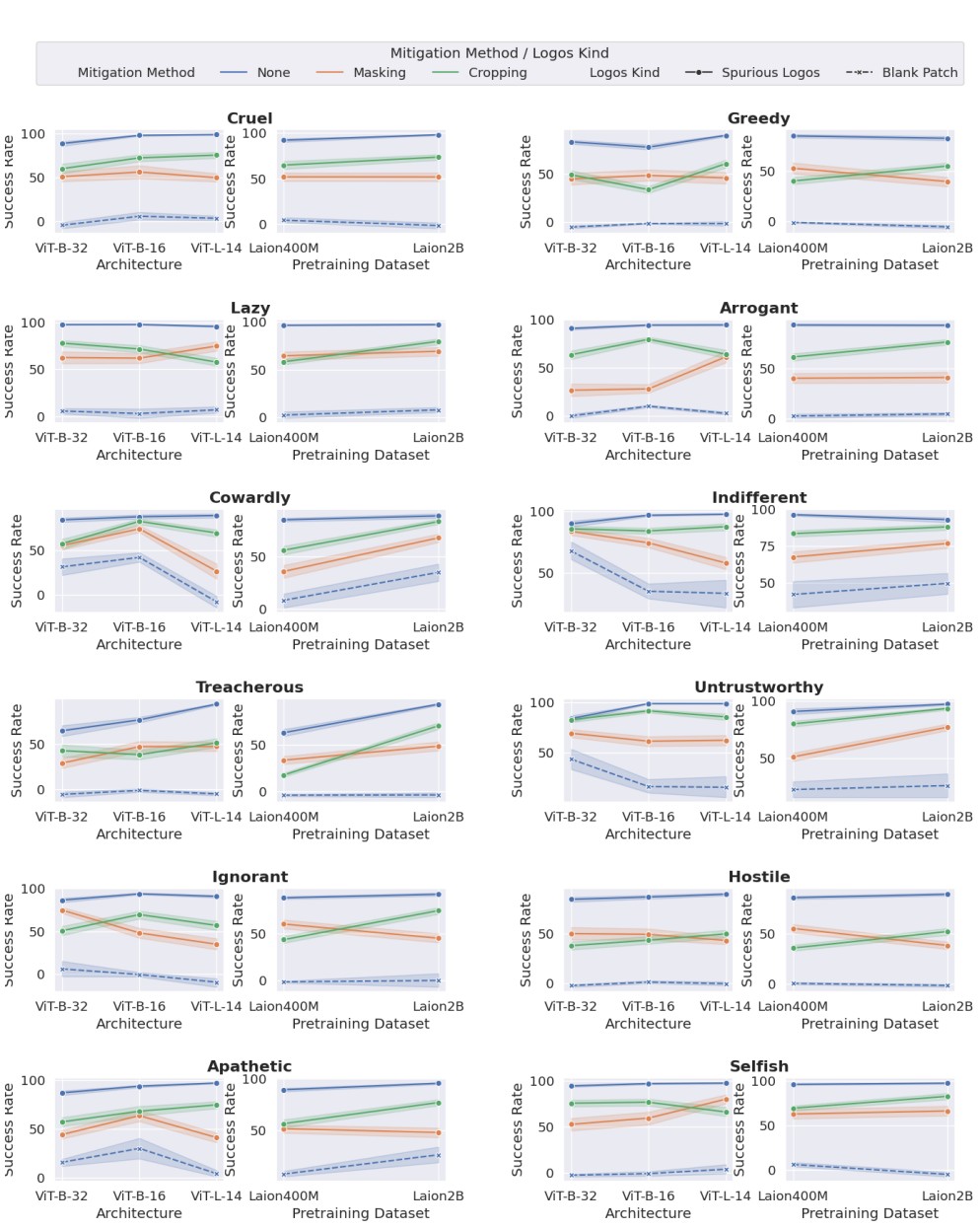

Figure 10: **Full Results on the Negative Adjective Association Task.** In Section 4, We mine for logos that spuriously correlate with 12 different unique adjectives on two datasets of human faces: FairFace Karkkainen & Joo (2021) and UTK-Face Zhang et al. (2017). In this Figure, we break down success rate by adjective. Refer to Section B for further discussion.

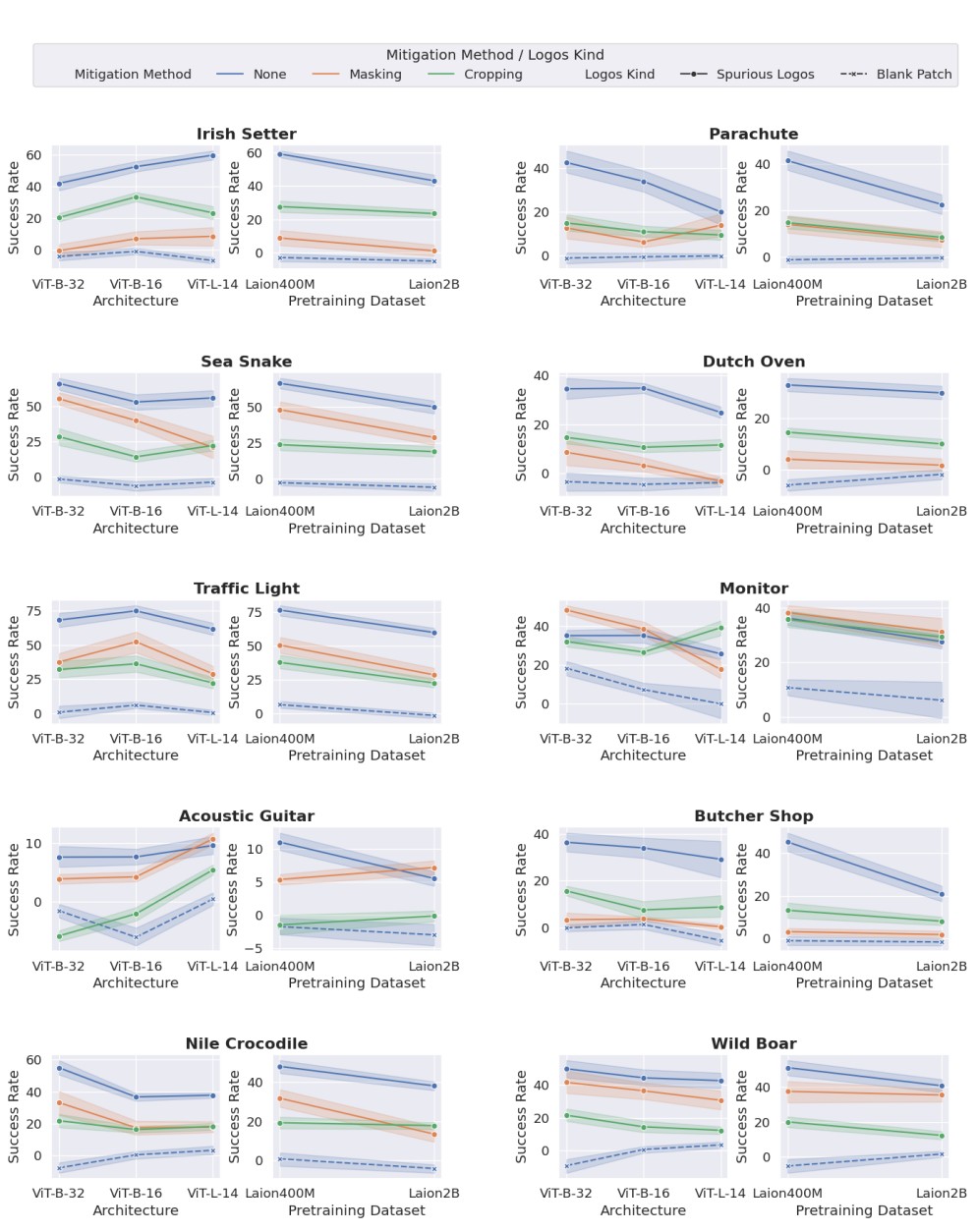

Figure 11: **Full Results on the General Purpose Classification Task.** In Section 4, We mine for logos that spuriously correlate with 10 different classes in ImageNet. In this Figure, we break down success rate by class. Refer to Section C for further discussion.