# OpenReview forum: "Hidden Logos in Web-Scale Data Disrupt Large Vision Language Models"
_ICLR.cc/2025/Conference — ICLR 2025 Conference Withdrawn Submission_

### Official Review · Reviewer_qXp5 · 2024-10-18

**Soundness:** 2
**Presentation:** 2
**Contribution:** 2
**Rating:** 3
**Confidence:** 4

**Summary:**

This paper investigates the spurious correlations between logos in images and the predictions made by CLIP. More specifically, it demonstrates that certain logos are strongly correlated with specific attributes and concepts. For example, adding a NASA logo to an image increases CLIP's prediction score for "harmless" content. The investigation introduces a subset of CC12M, named "CC12M-LogoBank," which was created by filtering the CC12M dataset for logos. Spurious correlations between logos and concepts are identified by measuring CLIP's performance on standard benchmarks and comparing its predictions for images with and without the logo of a particular company.

Experiments were conducted using three different CLIP architectures, each trained on two different datasets. The analysis examines the influence of model size, logo position, and the transferability of these correlations between models. Additionally, two mitigation strategies are explored: making predictions on augmented images (5-crop + flipping) and applying logo detection followed by masking.

**Strengths:**

- The phenomenon of spurious correlations of logos in CLIP models is interesting and contributes to the problem of spurious correlations and shortcut learning arising in models trained on large-scale, barely curated datasets like LAION. While the existence of spurious correlations in CLIP-like models is not surprising given the type of data they are trained on, the connection between logos and content moderation or ImageNet classes is not apparent at first glance. These connections identified by the paper are significant and call for improved data curation and robust training methods to mitigate such spurious correlations.
- Phrasing the effect from an adversarial perspective opens an appealing direction for discussion. While not explicitly mentioned by the paper, logos might act as some backdoor trigger or adversarial patch, thereby enabling an adversarial party to manipulate the prediction process without conspicuous changes.
- The paper proposed two practical solutions for mitigating the biases induced by including logos in the model inputs. While both methods add additional computation overhead, it is good to see practical mitigation strategies for this problem. However, using augmentations + majority voting as a mitigation strategy already exists in literature; see [1].

[1| Sarkar et atl, Backdoor Suppression in Neural Networks using Input Fuzzing and Majority Voting, IEEE Design \& Test

**Weaknesses:**

- The evaluation is limited to CLIP models. However, there exist multiple other VLMs, e.g., Llava, BLIP-2, LLama 3, etc. Other models should also be evaluated to support the hypothesis and findings that large-scale VLMs contain such spurious correlations between logos. Currently, this hypothesis is only empirically proofed for CLIP models, which are trained in a contrastive way. Other training methods might not lead to the same spurious correlations.
- The evaluation only uses a single metric (success rate of label swap). However, using relative metrics would provide more fine-grained insights into the biasing effect. For example, even if a logo does not change the predicted label, it might still sharply increase the model's confidence in the correlated class. Moreover, from an adversarial perspective, the label flip rate, counting the share of correct predictions that are corrupted by the logo, would be an interesting metric. Maybe check out metrics like the Relative Bias introduced by [4], which might be adjusted to this paper's setting and offer a more fine-grained evaluation.
- While phrasing spurious correlations from an adversarial perspective is intriguing, the paper should be better placed in the domain of adversarial attacks against ML systems, particularly in the context of backdoor attacks [2] and adversarial patches [3], since both concepts are closely related. Research in adversarial ML could also motivate additional evaluation metrics, such as label flipping rate, confidence reduction (untargeted adv. examples), etc.
- The paper explores no origins of this phenomenon. It would be interesting to see how, e.g., logos of sports clubs are represented in the training data to understand where their spurious correlations originate from. Probably, there is the tendency that certain logos are primarily placed with positive/negative captions.
- Overall, the contributions of the paper are limited. The effect of the spurious correlations between logos and unrelated concepts is intriguing, but the paper only provides a limited empirical evaluation of this phenomenon (see previous weaknesses). The paper introduces no novel general insights into the topic of spurious correlations (novel metrics, general mitigation methods, exploration of the origins, etc.), limiting its impact on this line of research.

Marginal Weaknesses / Opportunities to Improve
- Formally introduce the abbreviation VLM once. It already appears in the abstract, but once explicitly written, "Vision-Language Models (VLM)" would be clearer to the reader. Also, once introduced, use it consistently. In L116f, the paper again uses vision-language models as a term instead of its abbreviation.
- P in Eq. 2 should be defined once. It probably stands for the probability, but it is not entirely clear to me what exactly is measured here. Is it the average number of cases in which the model assigns the highest confidence to concept T (hard label prediction), or does the equation compare the average softmax probability scores for this concept?
- Instead of using line plots, bar charts would be a better visual representation given that there are only two points on the x-axis.

[2] Gu et al., BadNets: Identifying Vulnerabilities in the Machine Learning Model Supply Chain, Preprint (but highly cited)
[3] Brown et al., Adversarial Patch. NeurIPS 2017
[4] Struppek et al., Exploiting Cultural Biases via Homoglyphs in Text-to-Image Synthesis, JAIR 2023



--------------
Overall, I like the general direction of the paper and think there is much potential about it. So I encourage the author's to keep up their research, extend the experiments and metrics, and to put more efforts into exploring the reasons behind this phenomenon.

**Questions:**

- While the paper investigates three different CLIP ViT-architectures, what about ResNet-based CLIP models? What about other VLMs like LLava? Are these models similarly affected by logos?
- Instead of using 5-crop image augmentations as mitigations, could the same be realized by dropping vision tokens from the ViT during inference? If so, it would offer an architecture-specific mitigation strategy that could be realized right inside the model without the requirement to manually perform data augmentation.
- How does the visibility of the logos impact the success? If the logos are still successful in biasing the model's prediction when decreasing their visibility, the impact of the findings is stronger since the logos could then act as a hard-to-spot backdoor trigger. Similarly, how does the size of the logos impact the biases? Both questions could be answered by running experiments with varying the logos size or alpha value and measuring their impact as a function of the size/alpha value.
- Are the findings regarding the model sizes really due to the parameter count, or does the different patch size of the models impact their behavior? Using a smaller patch size would more likely cut the logos in pieces, whereas a larger patch size contains them. If possible, experiments using the same model architecture but with different patch sizes (+ adjusting the parameters required to work with these different patch sizes) might answer this question. However, I understand that training CLIP models is resource intensive and might not be realized easily. Maybe there are pre-trained models available on Hugging Face?
- What about using synthetic logos that are not present in the training data. For example, a diffusion model can be used to generate novel logos and evaluate their impact. It may not be about the logos themselves but rather their color, structure, etc. So generating novel logos with certain aspects (varying color, shape, size, ...) and repeating the experiments should answer this question.

---

### Official Review · Reviewer_zXvc · 2024-10-31

**Soundness:** 2
**Presentation:** 3
**Contribution:** 3
**Rating:** 3
**Confidence:** 4

**Summary:**

The paper explores the problem of hidden logos contributing to spurious correlations to vision-language models. It studies effect of the spurious correlations on content moderation, classification and negative associations. The paper finds an easy to reproduce spurious correlations in VLM that could be used for content moderation and classifications. The paper helps contribute towards conditions that need to be tested to make practical deployment and usage of such models more reliable.No evaluation in a commercial scenario.

**Strengths:**

- The paper explores an important problem of adding disclaimers to VLMs being used for flagging content.
- The paper also explores and finds easy to reproduce spurious correlations while bringing attention to an attack surface that could be used to bypass automated content moderators.
- The paper built a new logo bank for future research on this topic and also suggest using two straightforward techniques (cropping or masking them out) to prevent said attacks.

**Weaknesses:**

- Studying spurious correlations is important, but the paper fails to motivate the problem for VLMs.
- Lack of comprehensive ablations: The paper makes several design choices: logo-placement on borders, cropping logos, masking (thus introducing an empty patch), choosing top patches, using CLIP to evaluate. All such choices might be painting a picture that might need take careful curation.
- No limitations section.
- Though this study is a preliminary exploration into this area, directions on how to pursue it further would be appreciated. For example, how to detect and prevent spurious logo correlations when the attacker is a non-expert, in case of melding a logo into the image, such as milk cartons with chinese characters. Another area to give directions could be suggestions to training more robust VLMs instead of evading logos, as some logos and their occurrence is indeed informative (e.g., ISO standards, hallmarks, USDA organic).

**Questions:**

1. **Model Architecture Clarification**
   - Pointing to Section 4, ViT-L and ViT-B share the same architecture with different hyperparameter configurations
   - Please expand on why these specific ViT models were selected? Are these models being used anywhere for content moderation?
   - The ViT baselines have no information on how they came to be - are they pretrained?

2. **Experimental Design**
   - Logo application on top of the image is a secondary manipulation
   - I failed to see the actual effect in any practical scenario. For example, why would a smiling portrait (Fig. 7) have a handicap symbol? Most examples do not reflect any believable setting.
   - The authors need to consider cases where the logo was actually useful for identifying something
   - Are there any motivating numbers that aid in gauging the impact of such spurious correlations?

3. **CCM12M-LogoBank**
   - How many logos are there in CCM12M-LogoBank? 87k are images I guess
   - Is the logobank going to be released?
   - I guess the authors meant "web-datasets" and not "web-scaled datasets"
   - Expanding utility to include practical examples.

4. **Detection & Prevention**
   - Have authors considered using splice detection, such as methods based on Pixel value discontinuities, Lighting/shadow inconsistencies, JPEG compression artifacts, Edge irregularities? As the authors are claiming "non-expert" threat actors who would just paste a logo and pass it on. If that is the case, the logo would have boundary artifacts, which could be used to deactivate that area (here patch, as using ViT).
   - Fig 4(b) is not really needed, it's just two datapoints each, so n=2 samples. Also the results are inconclusive as the decrease/increase in success rate is minimal
   - Paper uses OWLv2 to help find objects. OWLv2 itself is a VLM - does it not get affected by the spurious correlations introduced by logos? The above splice detection part might help.

5. **Literature Review**: Adding related work where the VLMs are being used for content moderation and adjective association

6. **Technical Robustness**
   - Why not train something better that is learning more robust features rather than simply evading logo application? What if there is a light watermark that evades your technique?

Follow-up: What are the practical implications for current content moderation systems?

---

### Official Review · Reviewer_iYYr · 2024-11-02

**Soundness:** 2
**Presentation:** 3
**Contribution:** 2
**Rating:** 5
**Confidence:** 4

**Summary:**

This paper investigates the spurious correlations caused by logos in web-scale training data of LVLMs. The authors develop an approach to semi-automate the discovery of logos that lead to spurious correlations in VLMs on three tasks. Such spurious logos are used to create a new dataset. After that, the authors define a threat model that uses the logos as an adversarial patch to attack VLMs. Two mitigation strategies are studied, including cropping and masking.

**Strengths:**

### Originality
* The study of how logos can be a shortcut in VLMs due to web-scale training data is novel.
* The related works of spurious behavior of VLMs are well-discussed, including the behavior caused by other concepts similar to logos.
* The proposed new dataset is useful for future evaluation of a specific instance of shortcuts.

### Quality
* The methodology of mining spurious logos is generally solid. It verifies if a logo can become a shortcut over multiple models and source images, instead of just a single input.
* Overall the study in Section 3.1 is well executed and evaluated.
* The threat model of using logos as an attack is reasonable and easy to execute by attackers.

### Clarity
* The presentation is overall good and easy to follow.

### Significance
* As in the originality aspect, the proposed new dataset is useful for future evaluation of a specific instance of shortcuts.
* It's good to know that logos can be used in a targetted way to attack VLMs.

**Weaknesses:**

### Originality

**Q1: Novelty compared with prior works.**

Generally speaking, this paper studies a specific instance of logo-based spurious correlations. As Section 2 discussed, there are prior works studying other specific instances of shortcuts, such as the Chinese watermark. The main justification is that *"their technique requires a prior knowledge of the nature of possible spurious correlations"* and that this paper *"develops a semi-automatic process for uncovering spurious logos"*. The argument here can be improved. It's unclear to me what kind of "prior knowledge" is required by the prior work. Isn't knowing that logos can be one of the shortcuts also a prior knowledge?

### Quality

**Q2: False positives in the proposed dataset.**

In Section 3.1.1, the proposed dataset is created by setting the noise level (i.e., FPR, if I understand correctly) at 2%. It would be necessary and interesting to show some examples of such FPRs. More importantly, it seems that such FPs are not removed from the dataset. Does this mean the dataset contains 2% of data that is technically not spurious logos? If so, how would you eliminate it or justify the impact on future evaluations?

**Q3: The mitigation strategy is not robust and holds strong assumptions of the attacker.**

The cropping technique has several assumptions of the attacker: the logo is only added once to one of the four corners, the logo has a small size that will not appear in most of the crops, etc. Since the attacker can easily attach the logo to all four corners and the center, with different sizes, it would be hard for this technique to mitigate. I'm not very confident in this technique, given that cropping-based defense in the adversarial ML literature has been proven ineffective for a long time. Although the notion of adversarial noise does not apply to the threat model here, the fact that the attacker can arbitrarily transform (e.g., rescale, rotate, etc.) was not considered. Such transformations are potential evasion strategies of the proposed defense, including the masking one. Generally speaking, for a defense to be recognized as effective (or robust), a reasonable amount of effort is needed to show that it's hard for the attacker to evade the defense.

Another point missing (and worth discussing) is how the generic problem of shortcuts is mitigated in prior works. It would be interesting to see if there are any prior inference-time techniques that can be used for the logo setting.

**Q4: The study of logo-based spurious correlation as an attack can be deeper.**

This is a minor point, but I am slightly concerned that the current study of putting the logo at a few positions with a fixed size is ad-hoc and could not fully explore the potential of logo-based shortcuts. Thinking about the attack's perspective, the attacker can essentially put the logo at any location, any size, and any angle. This leads to the notion of a logo's robustness regarding the shortcut. This is somewhat similar to the "spuriousity metric" used in the paper, but the latter is unclear (see clarity) and only averaged over different data. Note that I'm not saying the study of logo-based shortcuts is insufficient; the evaluation is interesting, but framing it as an attack would need a deeper investigation.


### Clarity

**Q5: Regarding the spuriousity metric**

In L213-215, it's unclear how the "prediction rate of T" is computed -- what is the definition of $P(M(\cdot))$? I think the root problem here is that the output space of $M$ is not defined.

**Q6: Misc**
* L195, "noise level" may be confusing, a better term is FPR is I understand correctly.
* It seems that the last row in Figure 7 does not work. All logos have the same effect as the no-logo baseline.

### Significance

As mentioned above, I'm positive about the study's impact, but its current form may not be as impactful when fitting into the attack & defense perspective.

**Questions:**

See weaknesses.

---

### Official Review · Reviewer_c7Dc · 2024-11-02

**Soundness:** 2
**Presentation:** 2
**Contribution:** 2
**Rating:** 5
**Confidence:** 2

**Summary:**

This paper investigates the impact of logos in web-scale datasets on Vision-Language Models (VLMs). It highlights how logos can create spurious correlations, leading to incorrect predictions, such as associating the Adidas logo with the concept of "greedy." The authors introduce a new dataset, CC12M-LogoBank, and propose an algorithm to identify these spurious logos. They demonstrate the potential for logos to be used as attacks on VLMs and suggest two mitigation strategies: cropping and logo masking. The study underscores the need for further research to address these vulnerabilities in VLMs.

**Strengths:**

- **Comprehensive Analysis**: The paper provides a thorough investigation into the impact of logos on Vision-Language Models (VLMs), covering various tasks such as content moderation, object classification, and negative adjective association. This broad scope highlights the pervasive issue of spurious correlations in VLMs.
- **Novel Dataset**: The introduction of the CC12M-LogoBank dataset is a significant contribution. This curated collection of logos enables systematic study and experimentation, which can be valuable for future research in mitigating spurious correlations in VLMs.
**Potential Dataset Bias**: While I appreciate that the authors have included multiple datasets for evaluation, I am concerned about the potential overlap between these datasets and the VLM model's training data. If these datasets were part of the model’s training, how might this affect the reported results? And what if not? It would be beneficial if the authors could explore this point in greater depth.
- **Practical Mitigation Strategies**: The proposed mitigation methods, such as cropping and logo masking, are practical and integrate seamlessly with zero-shot inference of VLMs. These strategies show promise in reducing the spurious effects of logos, making the findings applicable in real-world scenarios.

**Weaknesses:**

- **Limited Scope of Logos**: While the study covers a range of logos, it primarily focuses on logos that are pasted onto images. This approach may overlook more complex scenarios where logos are naturally integrated into the visual context, potentially limiting the generalizability of the findings.
- **Why the three scenarios**: The authors consider three scenarios, i.e., content moderation, general purpose classification, and negative adjective association. Why consider these three scenarios? Are there containing other potential scenarios?
- **Potential Dataset Bias**: While I appreciate that the authors have included multiple datasets for evaluation, I am concerned about the potential overlap between these datasets and the VLM model's training data. If these datasets were part of the model’s training, how might this affect the reported results? And what if not? It would be beneficial if the authors could explore this point in greater depth.
- **Dynamic Nature of Logos**: The paper acknowledges that the associations between logos and visual concepts can change over time due to evolving marketing campaigns and public perceptions. However, it does not propose a method to address this dynamic nature, which could affect the long-term applicability of the findings.

**Questions:**

1. Does the study’s focus on logos pasted onto images limit its applicability to cases where logos are more naturally integrated into the visual context?

2. What was the rationale for selecting the three specific scenarios—content moderation, general-purpose classification, and negative adjective association? Are there other scenarios that might also be relevant to consider?

3. It would be beneficial if the authors could explore the effect of dataset bias.

4. How might the method adapt to the evolving associations between logos and visual concepts over time due to changing marketing campaigns and public perceptions?

---

### Note · Authors · 2024-11-14

I have read and agree with the venue's withdrawal policy on behalf of myself and my co-authors.